# Silver Nanoparticles Incorporated on Natural Clay as an Inhibitor against the New *ISO SS* Bacteria Isolated from Sewage Sludge, Involved in Malachite Green Dye Oxidation

**DOI:** 10.3390/molecules27185791

**Published:** 2022-09-07

**Authors:** Mirila Diana-Carmen, Raducanu Dumitra, Georgescu Ana-Maria, Rosu Ana-Maria, Ciubotariu Vlad Andrei, Zichil Valentin, Nistor Ileana-Denisa

**Affiliations:** 1Catalysis and Microporous Materials Laboratory, Department of Chemical and Food Engineering, Faculty of Engineering, “Vasile Alecsandri” University of Bacau, 157, Calea Mărăşeşti, 600115 Bacau, Romania.; 2Department of Biology, Ecology and Environmental Protection, Faculty of Science, “Vasile Alecsandri” University of Bacau, 157, Calea Mărăşeşti, 600115 Bacau, Romania; 3Department of Engineering and Management, Mechatronics, Faculty of Engineering, “Vasile Alecsandri” University of Bacau, 157, Calea Mărăşeşti, 600115 Bacau, Romania

**Keywords:** clay, silver catalyst, catalytic ozonation, dye, bacteria, antibacterial activity, water treatment

## Abstract

A facile, ecofriendly, and cost-effective method was developed to prepare a microporous material based on natural chemically modified bentonite with silver ions (BN-Ag^0^). This material presents a good catalytic activity against Malachite Green (MG) dye and bacteriostatic activity against a newly isolated bacterium from sewage sludge named hereafter “*ISO SS*” and *Escherichia coli* (*E. coli*). BN-Ag^0^ was characterized by the following methods: energy-dispersive X-ray spectroscopy (EDX), scanning electron microscopy (SEM), Brunauer–Emmett–Teller (BET), Fourier-transform infrared (FTIR) spectroscopy, temperature programmed desorption (TPD) and X-ray Diffraction (XRD). The new bacterium *ISO SS*, was isolated using the technique of isolating a pure culture of anaerobically stabilized sludge. A mandatory characterization of *ISO SS* isolated strains from anaerobic stabilized sludge was performed in the process of identifying bacterial species. The cationic clay-based nanomaterial showed appreciable antibacterial activity against *ISO SS*, a Gram-negative bacterium. It also showed good activity against *E. coli* bacteria. As a catalyst in the catalytic ozonation of MG dye, BN-Ag^0^ significantly improves the oxidation time of the dye, due to its good adsorption and catalytic properties. The catalytic and antibacterial activities of the natural bentonite (BN) and of BN-Ag^0^ were examined using performant characterization techniques. The lifetime of the BN-Ag^0^ catalyst was also evaluated. Results obtained are expected to provide valuable findings for the preparation of a good microporous material with multiple functionalities.

## 1. Introduction

Water pollution by organic pollutants, remains a major environmental issue to be addressed, because of the strong negative impact of such molecules due to both their direct toxicity and their derivatives. Water depollution has become one of the major challenges worldwide, because more than 50% of all countries will face this crisis by 2025 [1,2]. 

Among the wide variety of water pollutants, natural bacterial pathogens, organic dyestuffs and their derivatives are particularly hazardous for biodiversity and human health [3]. Primary water treatments that include coagulation–flocculation of organic matter is essential in remediation processes, but they often turn out to be quite insufficient for releasing clean water in nature or for other purposes [4]. In this regard, oxidative water treatments were found to be essential for complete remediation without any impact on biodiversity and human health. Urban wastewater treatment plants (WWTPs) are not designed to remove organic compounds, which is why many advanced oxidation processes (AOPs) have been investigated and implemented in WWTPs treatments. 

AOPs are widely used for the removal of recalcitrant organic constituents from industrial and municipal wastewater. These processes provide a variety of possible ways to produce hydroxyl radicals: by the direct use of ozone, or of ozone and hydrogen peroxide. The hydroxyl free radical (**•**OH) is even more reactive than chlorine, permanganate and hydrogen peroxide, and can rapidly react with a wide variety of organic substrates [5,6]. 

In the past few years, bacterial resistance to antibiotics has been rising and many infectious diseases have been reported [7], as a direct effect of antibiotic resistance [8]. Though nanoparticles of gold, zinc and copper have showed high bioactivity, silver nanoparticles are the most promising. As reported [9,10], silver ions released from unstable silver nanoparticles (AgNPs) are responsible for the excellent biocide action. AgNPs have been extensively employed in different healthcare products such as: antiseptic sprays, wound dressing bandages and dermatological creams. The ability of AgNPs to disrupt pathogen membranes, leads to impairing the microorganisms’ enzymatic activities [11]. AgNPs are also used in cleaning agents, textile coatings, food packing and storage and medical devices [12]. Excessive use of antibiotics has led to the resistance of microorganisms to them, thus becoming a growing health problem. In the literature, several authors [10,13] have demonstrated that due to their small size and large surface area to volume ratio, AgNPs have potent antimicrobial activities against bacterial strains.

The presence of dyes in the environment has a negative impact on biodiversity, especially on aquatic fauna. Due to this inconvenience, the authorities should impose a stricter regulation on industrial producers to remove dye-rich wastewater [14]. The special interest dedicated to organic dyes results from their much higher concentration compared to a wide variety of organic pollutants released in wastewater from different industries (food, pharmaceutical, textile etc.), hospital effluents and others [15,16,17,18].

MG (also known as Diamond Green B or Victoria Green B) is classified in the dye industry as a triarylmethane dye. It is usually used in agriculture and the fishing industry as a powerful anti-bacterial, anti-fungal, anti-parasitic and dye agent, but it is also used for paper, leather, silk and other materials [18]. The use of MG has been forbidden in several countries as it is not approved by the US Food and Drug Administration, but this dye is still widely used due to its low cost, easy availability and efficacy [19,20]. MG possibly causes carcinogenic, mutagenic and teratogenic effects on humans, if it enters the food chain [18]. It is also highly cytotoxic to mammalian cells and acts as a tumor enhancing agent [21,22].

Naturally occurring bacteria do not develop resistance to AgNPs, as already reported in the literature [23]. Therefore, a special interest was devoted to Ag nanoparticles. In this regard, we developed a low-cost mechanism to synthesize a promising BN-Ag^0^ clay material.

To our knowledge, it is the first time that the natural sodium bentonite from Valea Chioarului, Maramureș (Romania), was used as raw material for the synthesis of chemically modified bentonite with silver nanoparticles. Indigenous bentonite, as an active phase support, is an alternative to activated carbon, zeolites or other materials, due to its availability in nature, good adsorption properties and very low price of exploitation [24]. BN intercalated with silver oxide was prepared, followed by the chemical reduction of silver with sodium tetra hydroborate, named BN-Ag^0^. The chemically modified material has been evaluated against the development of a new isolated bacterium from sewage sludge, named *ISO SS*, which was provided from the sewage treatment plant in Bacau, Romania. In addition to bacteriostatic/bactericidal activity, the catalytic activity of this material was evaluated for the total mineralization of MG dye. 

To our knowledge, no studies have investigated the functionalization of BN-Ag^0^ catalyst that removes MG dye.

## 2. Results

### 2.1. Experimental Protocol for the Synthesis of BN-Ag^0^

The experimental protocol for obtaining the chemically modified natural cationic clay consisted of two steps. In the first step, the bentonite was ion exchanged once. For this purpose, 2 g of bentonite was treated with 2 M NaCl aqueous solution under vigorous stirring at 60 °C. The resulting material was repeatedly washed with deionized water, filtered through centrifugation, and then dialyzed through a cellulose membrane until a chloride-free supernatant was obtained. The filtered powder was further dried at a temperature of 80 °C for 2 h. In the second step, 2 g of Na-exchanged bentonite was dispersed in distilled water (2 wt% slurry) and the suspension was stirred for 1 h, at room temperature. Further, 50 mL of 0.2 N AgNO_3_ solution was added to this slurry, drop by drop, under vigorous stirring and mixed for 10 min, at room temperature. A total of 10 drops of 10% NaBH_4_ solution was added to this slurry, to reduce the ions of Ag^+^ to Ag^0^, under vigorous magnetic stirring for 1 h, at 50 °C. The obtained solution was filtered with double distilled water for NO_3_¯ ion removal. The obtained material was dried for 2 h at 120 °C.

### 2.2. Material Characterization

The raw material used for the synthesis of BN-Ag^0^ catalyst was natural BN. Its mineralogical composition is presented in Table 1. BN raw material has a high content of montmorillonite. For this reason, it is a suitable candidate for obtaining adsorbents with catalytic activity that can be effective in destruction of industrial pollutants.

Natural and chemically modified clays were characterized by the following analysis: EDX, SEM, BET, TPD, XRD and FTIR. 

The particle morphology of the surface was investigated by scanning electron microscopy which is presented in Figure 1.

The morphological analysis of the raw material (Figure 1a,c,e,g,i) reveals the presence of large, clearer particles, which may correspond to the presence of quartz, highlighted by DRX analysis. The morphological analysis of BN-Ag^0^ (Figure 1b,d,f,h,j) shows the presence of small clearer particles, with smooth surfaces. The surface morphology of the modified bentonite is different from the raw material, by presenting grain-like particles with a fluffy appearance, revealing its extremely fine plate-like structure. The surface has become more porous by the binding of Ag^+^ ions in the interlayers of clay. A decrease in the particle size of the BN-Ag^0^ is clearly visible, the nanoparticles having the tendency to form aggregates. 

The results obtained by EDX analysis in six distinct points, for the raw material, respectively for the chemically modified bentonite presented in Figure 2. 

Incorporation of Ag^+^ ions was confirmed by EDX. Ag was found in all six points analyzed. Its weight was in the range of 0.5–2.9%. The second area analyzed presented the largest amount, followed by area one with 2.2%.

The pore size distribution curves of indigenous natural clay and of chemically modified clay are presented in Figure 3. The natural clay contains mainly meso and micropores, this variation could be explained by the presence in the structure of different minerals with different characteristics. The chemically modified natural clay (Figure 3b) also contains meso and micropores in its structure. The increase in height of the peaks characteristic of the distribution curves in the mesoporous region highlights the formation of a large number of mesopores (2–50 nm).

In Figure 4, the isotherms obtained for BN compared to BN-Ag^0^ are shown. They are similar in shape. According to IUPAC classification [25] of adsorption isotherms, the isotherms obtained are similar to those of type IV and show a hysteresis corresponding to the formation of pore aggregates in the form of slits and of variable sizes.

The specific surface areas of the natural clay and of chemically modified clay BN-Ag^0^ are presented in Table 1. By chemical modification, the specific surface area of the clay decreases from 25.80 m^2^/g to 24.29 m^2^/g according to Table 2.

The values of total surface acidity for BN and BN-Ag^0^ obtained and presented in Table 2, are in accordance with the theories developed in the literature, the surface acidity being influenced by the number of protons in the interleaving solution. 

The NH_3_-TPD profiles for BN and BN-Ag^0^ are shown in Figure 5. From this figure, it can be determined that NH_3_ molecules are now fixed on active material sites. From the profile of the forks, it can be concluded that between the acidic active sites of the material and the weakly basic NH_3_ molecules, weak bonds have been created that can break relatively easily when the temperature increases. This is also the interest pursued to be able to use the materials in adsorption–desorption processes. The incorporation of Ag probably also led to an increase in the acidity of the material. The structural changes during preparation lead to the partial or total destruction of the octahedral layers in the crystalline structure, which immediately results in an increase in Lewis acidity—attributed to surface cations. During the preparation, in order to establish the balance at the charge level, the protons in the ion exchange solution replace the exchange interlamellar cations. These protons contribute to increasing the surface acidity. It is also possible that the protons in the marginal -OH groups of the octahedra become more labile due to structural deformations due to ion exchange, which also leads to an increase in Brønsted acidity. In the case of the adsorption of basic gases (ammonia), it is important that the adsorbent has a surface acidity as high as possible for the best efficiency of the adsorption process. 

The mineralogical data obtained and presented in Figure 6, are similar to other results published in the literature for exploited natural bentonites [26], being composed mainly of montmorillonite and non-clay minerals such as cristobalite and quartz. The results of the XRD analysis indicate that the crystallinity of BN increases with chemical modification. According to the XRD diffractograms shown in Figure 6, the presence of silver in the clay structure can be observed at the value of 2 θ of 27°. According to the XRD analysis, the mineralogical composition of BN-Ag^0^ is presented in Table 3.

The crystal lattice of the synthesized material is not destroyed, it is partially preserved, and the non-clay fractions do not change.

FTIR analysis reveals the differences that occur in the chemical structure of clays after the chemical modification process. Depending on the variations in the wave number, the free cations as well as the cations in the interlamellar space are highlighted. They occur due to interactions that took place in the clay matrix during the chemical modification process. 

In Figure 7, the FTIR spectra corresponding to BN and to BN-Ag^0^ are presented. It is observed that the FTIR spectra contain bands characteristic of smectite clays. The intense bands that appear in the spectrum are assigned to the following groups of molecules: 3735 cm^−1^ corresponds to AlAlOH vibration coupled with AlMgOH and SiO-H vibration; 3655 and 1650 cm^−1^ correspond to the interlamellar water absorption bands (H-O-H); 1000 cm^−1^ corresponds to the elongation vibration of -Si-O; 800 cm^−1^ corresponds to -Si-O vibrations in various modified forms of silica, Si-O-Al vibrations in lamellar silicates and (Al, Mg) -O-H; 475 cm^−1^ corresponds to -Si-O-Al and Si-O-Mg coupled with OH vibrations or Si-O vibrations.

Ag incorporation was confirmed by EDX, DRX as previously presented, and FTIR analysis which revealed the appearance of a new bands at 480–510 cm^−1^, 780–805 cm^−1^ and 1050–1060 cm^−1^.

### 2.3. Bacterium Characterization 

The characterization of the *ISO SS* strain was performed by Gram staining, colony cultural characteristics (appearance, shape, color), oxidase test (Table 4) and spectral analysis using FTIR technique. Recording spectra obtained from isolated bacteria in a usual nutrient agar medium, was accomplished by transferring a portion of the colony with sterile instruments, to avoid contamination of the samples. 

The structural and compositional characteristics of isolated pure bacterial strains derived from the anaerobic stabilized sludge were identified using the FTIR technique. It is known that the cell membrane spectra contain the main modes of vibration of lipopolysaccharides and proteins. Each spectrum recorded for isolated strains was mathematically processed (the second derivative was used, which allows the highlighting of “hidden” maxima) [27]. In Figure 8 it can be observed that in the spectra for the strain isolated in the 900–1530 cm^−1^ range, the characteristic bands of the sludge and of the nutrient medium (Gelose) are not found, as presented in (Table 5). 

For the correct evaluation of the spectral data, additional untreated sludge samples (N) as well as additional sterilized sludge samples (NS) were subjected to analysis (Figure 9a). The second order derivatives of Geloza G nutrient medium (Figure 9b), of N (Figure 9c) and of NS (Figure 9d) were obtained following additional sterilization by autoclaving, respecting standardized procedures.

Figure 9a shows that the ATR spectrum for the NS slurry differs substantially from the spectrum of the N sludge (in which the microorganisms are viable) in the spectral range of 2000–550 cm^−1^. This finding is confirmed by the derivation of the second order derivative of the original spectra of gelatin, N and of the additional sterilized NS by autoclaving. Therefore, it can be argued that in the spectral range of 2000–550 cm^−1^ the differences between the spectra of each bacterial strain investigated can be identified. The spectral range chosen is different from the one used in the literature [32].

Figure 9b shows major differences between the N-sludge spectra and the extra treated N-autoclave in two distinct domains 550–875 cm^−1^ (I) and respectively 990–633 cm^−1^ (II). Since the fingerprint domain is considered by most authors [29,32,33] to be attributed to the range of 1800–550 cm^−1^, we propose to analyze in more detail the spectra of isolated *ISO SS* in this spectral domain. In the first domain (I), the bands are mainly caused by carbon bonds. The maxima in the region of 586 cm^−1^ corresponds to the deformation vibrations of the CH bonds outside the plane. The second domain II, specific for N and NS sludge (Figure 9c,d), has two particularities at 1537 cm^−1^ and 1560 cm^−1^, which correspond to the Amide II groups. 

The ATR spectra were studied in the 550–1800 cm^−1^ range, in order to characterize isolated bacterial strain in terms of molecular structure. The purpose of these analyzes was to identify the fingerprint of the bacterial strain that remained resistant to the applied treatment. Because in the 550–800 cm^−1^ range, there were no well-defined particularities, the field of investigation was reduced to 900–1700 cm^−1^. Because the results of the investigations of isolated peculiar strains do not interfere with the nutritive medium on which the isolated strains were grown, the spectral characteristics of G-gel (nutrient medium for isolated strains) were also identified (Figure 9b). The assignment of the main FTIR absorption bands for nutrient agar (G) medium, anaerobic stabilized treatment sludge (N) and anaerobically stabilized anaerobic treatment sludge by autoclaving (NS) is synthesized in Table 6. The assignment of the main ATR absorption bands of the agar nutrient (G) is used only to avoid possible overlapping of the bands.

### 2.4. Antibacterial Activity

The results obtained showed a response for all tested samples, as can be seen in Figure 10. An inhibition zone was observed, and bacteria don’t remain on the entire surface, showing the presence of an inhibition zone with a diffusion process around the sample. The diameter of the inhibition zone (DIZ) against *ISO SS* using clay incorporated with Ag^+^ ions (Table 7) was found to vary according to the amount of the material. 

Ceftaroline fosamil CPS 30 and ciprofloxacin CIP 5 antibiotics were used to test the resistance of *ISO SS* and *E. coli* bacteria to antibiotics. As expected, *ISO SS* did not present antibiotic resistance, as can be seen in Figure 11, as well as *E. coli*. However, better results have been obtained with the use of ciprofloxacin CIP 5 antibiotic for both bacteria.

### 2.5. Catalytic Activity

In parallel with its antibacterial activity, the catalytic effect of BN-Ag^0^ on MG was evaluated. To correctly evaluate the improved catalytic activity of the BN-Ag^0^ nanomaterial, the catalytic activity of BN was also evaluated.

#### 2.5.1. The Effect of Catalyst and Ozone Dose

In Figure 12 are presented the catalytic ozonation processes in 20 mL of MG dye solution (5 × 10^−5^ M), using different amounts of BN and BN-Ag^0^ (2.5, 5, 10 and 15 mg) and different doses of O_3_ (0.5 and 1 g∙h^−1^), the ozonation time being between 0–300 s.

As can be seen in Figure 12c, the best yield in the degradation of MG dye was 86.35% when 2.5 mg/20 mL of BN-Ag^0^ and 0.5 g/h of ozone was used. Compared with the simple ozonation of MG published in a previous work [18] and catalytic ozonation with BN, the introduction of the BN-Ag^0^ greatly reduces the duration of the process, from 1 h and 50 min in just 300 s with a yield of 86.35%. The catalytic effect of material is probably due to its capacity to define a porous surface and produce acidic sites. This effect is evidenced by the evolution over time of the difference between the relative absorbance of the sample with and without the catalyst, expressed as (A/A_o_).

#### 2.5.2. The Effect of pH during Catalytic Ozonation

It was considered that the pH of the reaction mixture evolves differently over time, depending on the nature of the dye, the ozone flow, ozonation time, the presence of the catalyst and its type. The effect of the pH of the MG dye solution (5 × 10^−5^ M) during the ozonation process was published in a previous paper [17]. The introduction of the catalyst (BN or BN-Ag^0^) into the dye solution changes the pH value (from 6.5 to 8.5 and 6.5 to 8, respectively), as is presented in Figure 13. There are several reasons for this effect, but the most important is the presence of mineral impurities in the catalyst (cristobalite and quartz). Another reason is represented by the initial pH of reaction. In the experiment of MG degradation, the augmentation of pH led to a small decrease in catalytic activity. This effect can also be explicated by the positioning of electrical charges on the lamellae layers of the catalytic material.

The change in pH during catalytic ozonation with BN-Ag^0^ may be also due to the slightly basic increase in amphoteric hydroxylated intermediates, in accordance with the literature [6,36].

#### 2.5.3. Catalyst Recyclability

The BN-Ag^0^ catalyst can be recovered and reused in repetitive cycles of destruction. After each cycle, the catalyst was recovered by simple centrifugation and then washed with double-distilled water for the following cycles of heterogeneous catalysis. The reused catalyst involves two factors: the recyclability by complete recovery of the catalyst from the reaction mixture and the stability of its catalytic activity. It was found that the stability of its catalytic activity decreases from 86.35% to 73.34% after six cycles of ozonation for the destruction of industrial pollutant, MG (Figure 14). This can be considered as an appreciable stability for potential use in practical applications in industry. Research is still ongoing in this direction.

The results reported in Figure 14, indicate that the catalytic material, BN-Ag^0^ can be recycled and reused more than six times without degradation of its catalytic activity on MG, below 73%. When discussing the reuse of the material for repeating the process, the possibility of complete recovery of the catalytic material from the mass of reagents and the preservation of its catalytic activity must be considered. From the experiments undertaken, it was found that after six cycles, the catalytic activity is reasonable, which leads us to the conclusion that it is efficient to use this material. In repeated ozonation tests of MG, it was found that the catalytic activity decreases slightly. This fact is possibly due to the denaturation of the active sites of the material. Also, Ag^+^ ions can contribute to the reduction in the catalytic activity due to the transformation undergone in the oxidation process with ozone.

## 3. Discussion Regarding the Damage of *ISO SS* Bacteria Cell 

To explain how the damage of bacterial cells takes place, upon contact with BN-Ag^0^ nanoparticles, a possible mechanism has been proposed to describe the antibacterial behavior, as follows. After treatment with BN-Ag^0^, the amount of *E. coli* greatly decreased, where damage and leakage of intracellular contents could be occurring. The nature of the silver ions may be one of the possible reasons for the inhibition of bacterial growth. A possible explanation of the antibacterial activity could consist of a partial AgNO_3_ dissolution and release of Ag^+^ cations. Another idea is that silver ions interact with the thiol groups present in proteins of the cytoplasm, causing denaturation of proteins, which impairs replicating ability. Because *ISO SS* cell walls have a net negative charge, an electrostatic attraction is created between silver ions and the cell wall in which the functional groups (such as: carboxyl, phosphate, and hydroxyl) strongly bind to silver ions via ion-dipole interactions, causing shrinkage of the cytoplasm membrane or detachment of the cell wall. Because the silver nanoparticles are not uniform in shape and do not have a narrow particle size distribution [1], the material tested in this study presents an appreciable performance in terms of antibacterial activity.

## 4. Materials and Methods

### 4.1. Materials

All reagents used in this study were purchased from Sigma-Aldrich (St. Louis, MO, USA): silver nitrate (AgNO_3_), sodium chloride (NaCl), sodium tetra hydroborate (NaBH_4_), ethanol (C_2_H_5_OH), Malachite Green (MG, 0.05 wt.% in H_2_O, C_23_H_25_N_2_, MW: 364.911 g∙mol^−1^), Levenhuk immersion oil (gram staining kit for microscopy). BN was mined from the Chioarului Valley, Maramures County, Romania. Cefazoline fosamil CPS 30 (C_22_H_21_N_8_O_8_PS_4_) and ciprofloxacin CIP 5 (C_17_H_18_FN_3_O_3_) antibiotics were obtained from Oxoid (St. Louis, MO, USA). *E. coli* (ATCC 25922) strain purchased from Thermo-Scientific (USA). MG, a cationic dye, was used as a probe molecule in aqueous 10^−5^ M solutions. Double distilled water was used throughout this work.

### 4.2. Inoculation of the Bacteria

The inoculums were the *ISO SS* bacterial suspensions with a density of 0.5 McFarland [37] from a young culture with 18–24 h of incubation at 37 °C. Inoculation was performed with 600 μL bacterial inoculums and uniformly displayed on Muller–Hinton medium. The Petri dishes were left 3–5 min on the worktable to absorb the inoculums of the culture medium. The natural chemically modified bentonite with silver ions was added over the inoculums in a round shape in different doses (10 to 30 mg), and was then moistened with 100 μL ultrapure water. The experiment was repeated three times for each experimental variant. The Petri dishes were placed in the thermostat at 37 °C. After 24 and 48 h, the inhibition zones (in mm) were read, by observing the action of bacterial growth inhibition around the BN-Ag^0^. Bactericidal/bacteriostatic effect testing was based on the Kirby–Bauer model.

### 4.3. Antimicrobial Behavior of Ceftaroline Fosamil CPS 30, Ciprofloxacin CIP 5 and BN-Ag^0^ against ISO SS and E. coli

Ceftaroline fosamil CPS 30 and ciprofloxacin CIP 5 antibiotics purchased from Oxoid (USA) were used to test the resistance of *ISO SS* bacteria to antibiotics. As expected, *ISO SS* did not display antibiotic resistance. However, better results have been obtained with the use of the antibiotic ciprofloxacin CIP 5 in the case of *ISO SS*. The antimicrobial behavior of BN-Ag^0^ nanoparticles was evaluated by means of diffusivity and zone inhibitory tests against Gram-negative *ISO SS* bacterial strains isolated from the sewage sludge and the Gram-negative bacterial strain, *E. coli.* The test is a semi-quantitative method where the chemically modified material was directly contacted with a bacterial suspension spread on Muller–Hinton agar plates. After 24 h of incubation at 37 °C, the inhibition zones and diffusion zones of the material with silver ions were observed and analyzed. Different amounts of clay material were used (from 10 to 30 mg).

### 4.4. Catalytic Ozonation Working Protocol

Various doses of catalyst (2.5, 5, and 15 mg/20 mL) and of O_3_ (0.5 and 1 g∙h^−1^) were used in the catalytic ozonation process of MG. The ozonation time was in the range of 0–300 s. The addition of the clay-based catalyst should lead to the catalytic destruction of this dye in the presence of ozone. The first step is to remove the chromophore groups by breaking the MG molecule into smaller and less toxic molecules, following the oxidation of benzene by hydroxyl radicals to 2,4-hexadiene-1,6-dione. If there are enough •OH radicals in the reaction, subsequent attacks on 2,4-hexadiene-1,6-dione will continue until the fragments are all converted into small and stable molecules like H_2_O and CO_2_. This statement is in agreement with a previously published paper [22].

### 4.5. Devices

To be used in catalytic processes, it is important to know the characteristics of the chemically modified clay-based materials. For the structural characterization of BN, the specific surface area was determined using the BET (Brunauer–Emmett–Teller) method, the pore size distribution by Barrett–Joyner–Halenda (BJH) calculation and nitrogen adsorption–desorption isotherms. Prior to the adsorption measurements, the samples were degassed at 160 °C under vacuum for 4 h. The nitrogen adsorption–desorption isotherms were recorded at 77.35 K in the relative pressure range P_s_/P_0_ = 0.005–1.0, using a NOVA 2200e Gas Sorption Analyzer (Quanta chrome). Data processing was performed using Nova Win software, version 11.03 (Quantachrome Instruments, Boynton Beach, FL, USA). 

The total surface acidity of BN and BN-Ag^0^ was determined by the temperature programmed desorption method. Surface acidity is expressed by the amount of desorbed gaseous NH_3_ in the temperature range 150–400 °C.

X-ray diffraction was used to analyze the changes that occur in the crystalline structure of chemically modified clays. The analysis was performed using a Rigaku Geigerflex diffractometer. This device is operating at 50 kV and 40 mA and with a CuKα radiation λ = 0.157 nm.

FTIR spectra were recorded using an Agilent Technologies Cary 630 FTIR spectrometer. Measurements were performed in the spectral range of 550–4000 cm^−1^.

For the bacterium characterization, the total attenuated ATR reflection of the Tensor 27 infrared absorption spectrophotometer, Bruker, was used. Spectra were recorded on the 4000–550 cm^−1^ range, with a resolution of 4 cm^−1^ and 36 scans. Prior to the recording of each sample (obtained in triplicate) the air spectrum (background) was recorded. The spectral data obtained was then processed with ORIGIN version 9 software. 

A laboratory ozone generator (OZONFIX, Romania) was used to produce different concentrations of O_3_ from ambient air. Ozonation was carried out in a cylindrical glass reactor of 30 mL capacity, by bubbling ozone-air mixtures with different ozone concentrations (0.5 and 1 g∙h^−1^). The experiments were run at room temperature (25 °C). Samples of MG solutions (10 mL) were taken at regular time intervals (30 s), centrifuged to remove the solid (3000 rpm) and analyzed with a UV–Vis HELIOS OMEGA spectrophotometer (1 cm quartz cell). 

The acidity and basicity measurements were carried out by temperature programmed desorption of ammonia or carbon dioxide. The plant works under a nitrogen flow, the basic or acidic gas being introduced during experiments to study the acidic or basic properties of the material. 

In the work protocol, the speed of achieving the thermal regime was programmed to 2 °C·min^−1^. A granulometric fraction of 0.02–0.2 mm particle diameter was selected from the solid sample. Approximately 0.2 g of sample was introduced into the reactor. The sample introduced into the reactor is thermally treated at 400 °C for 4 h under a flow of N_2_. Then the sample is brought to a temperature of 120 °C for acidity control and about 80 °C for basicity. At a temperature of 120 °C, NH_3_ was injected for 5 min, then a nitrogen wash for 2 h was used to remove ammonia not bound by physical–chemical bonds (weak bonds) to the material. The desorption of NH_3_ was carried out by raising the temperature in the reactor. There was a direct link between the amount of desorbed ammonia and the active acid centers of the material (a large amount of desorbed ammonia shows the existence of many active acid centers). To measure the basicity, the procedure is similar, but CO_2_ is injected.

The bentonite samples (BN and BN-Ag^0^) were deposited on carbon tape and coated with 6 nm of platinum using the Leica EM ACE200 Sputter coater. The SEM analysis was done using a Verios G4 UC microscope (Thermo Scientific, Czech Republic, Prague), with a secondary electron detector (Everhart–Thornley detector, ETD), at a voltage of 5 kV. The microscope was coupled to the EDX mode, using the Octane Elect Super SDD detector, Mahwah, NJ, USA.

## 5. Conclusions

The best yield in the degradation of MG dye was 86.35%, being obtained by using 2.5 mg/20 mL of BN-Ag^0^, 0.5 g/h of ozone and an ozonation time of 300 s. Compared with the simple ozonation and catalytic ozonation with BN, the presence of the BN-Ag^0^ greatly reduced the duration of the process, from almost 2 h to just 5 min. BN-Ag^0^ material can be recycled and reused more than six times without degradation of its catalytic activity on MG dye (below 73%).

In parallel with the catalytic degradation of MG, the antibacterial activity of BN-Ag^0^ was evaluated for the *ISO SS* strain. An inhibition zone was observed, and bacteria did not remain on the entire surface, showing the presence of a 4 mm zone of inhibition with a diffusion process around the sample. The results obtained were in agreement with the literature [23], since the *ISO SS* bacterial strain did not develop resistance to BN-Ag^0^. Similar results were obtained in the case of the *E. coli* strain.

The recycling analysis of the synthesized nanocomposite showed that the BN-Ag^0^ is stable even after six cycles with a minor change in degradation.

The studied nanomaterial (BN-Ag^0^) presents interesting properties both for the oxidative degradation of MG-type dyes and for a wider use due to its antibacterial properties.

This material has a double role: as a material with a catalytic character as well as an inhibitory material for the development of Gram-negative bacteria. 

It can be concluded that the BN-Ag^0^ material is a nanomaterial with good textural, structural and morphological properties, which was chosen with the aim of using it in MG retention/destruction from aqueous solutions.

## Figures and Tables

**Figure 1 molecules-27-05791-f001:**
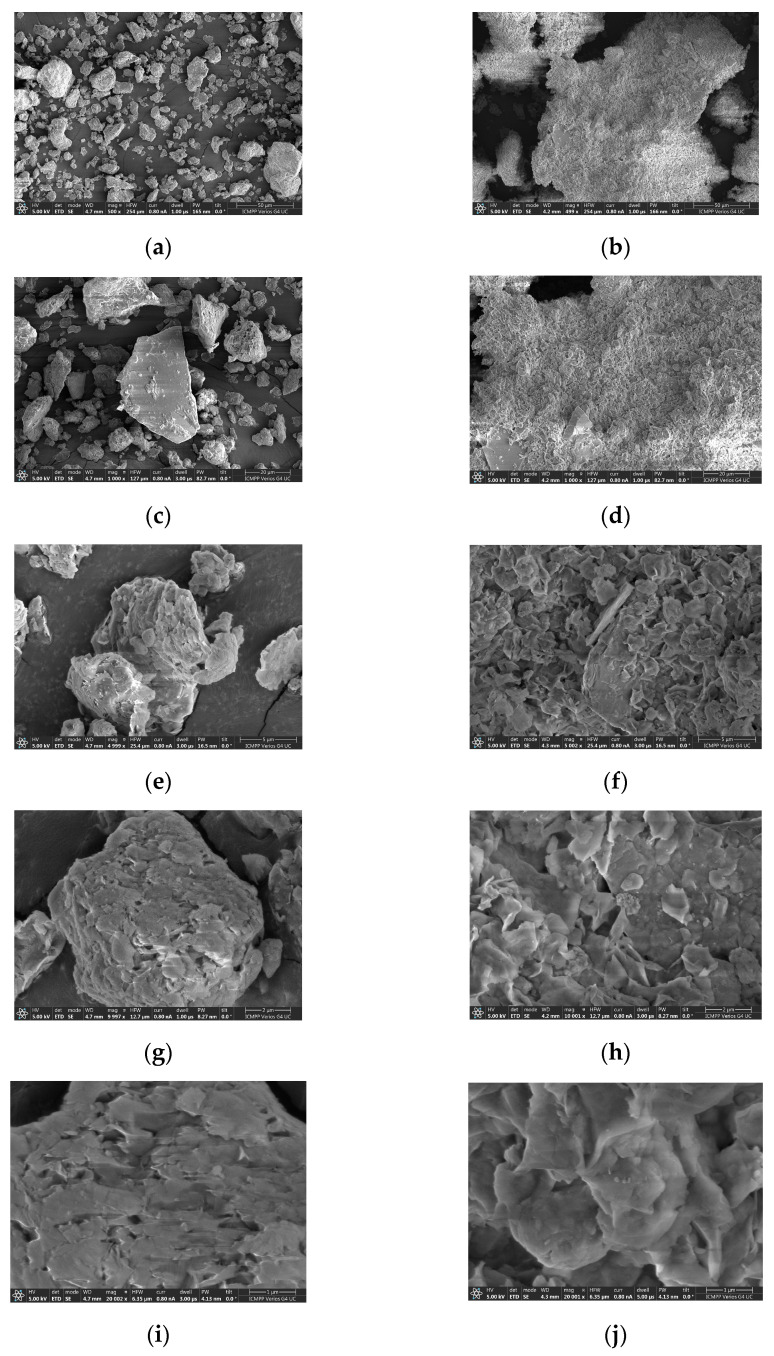
SEM analysis for BN (**a**,**c**,**e**,**g**,**i**), and for BN-Ag^0^ (**b**,**d**,**f**,**h**,**j**) samples at different units scale: 50 µm (**a**,**b**), 20 µm (**c**,**d**), 5 µm (**e**,**f**), 2 µm (**g**,**h**) and 1 µm (**i**,**j**).

**Figure 2 molecules-27-05791-f002:**
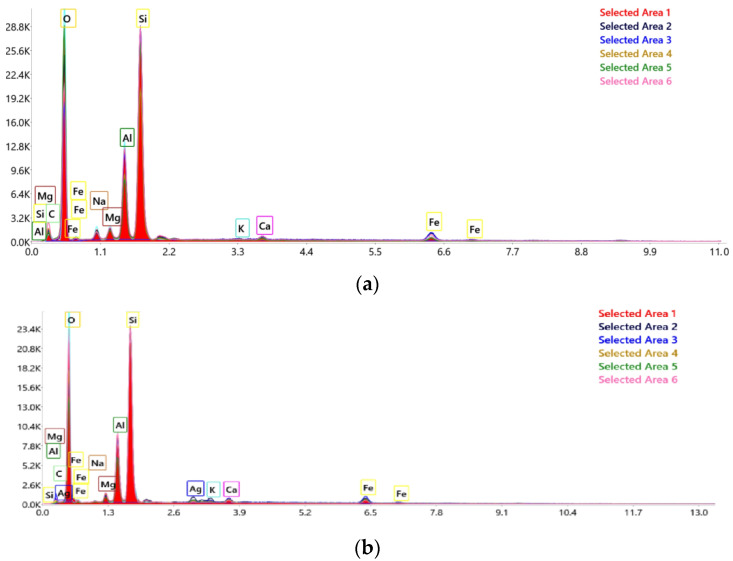
EDAX analysis for six distinct points for BN (**a**) and for BN-Ag^0^ (**b**) samples.

**Figure 3 molecules-27-05791-f003:**
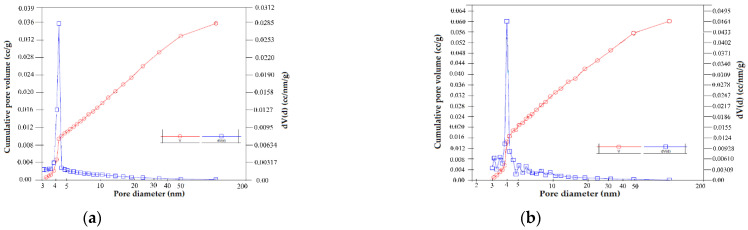
Pore size distribution curves of BN (**a**) and BN-Ag^0^ (**b**).

**Figure 4 molecules-27-05791-f004:**
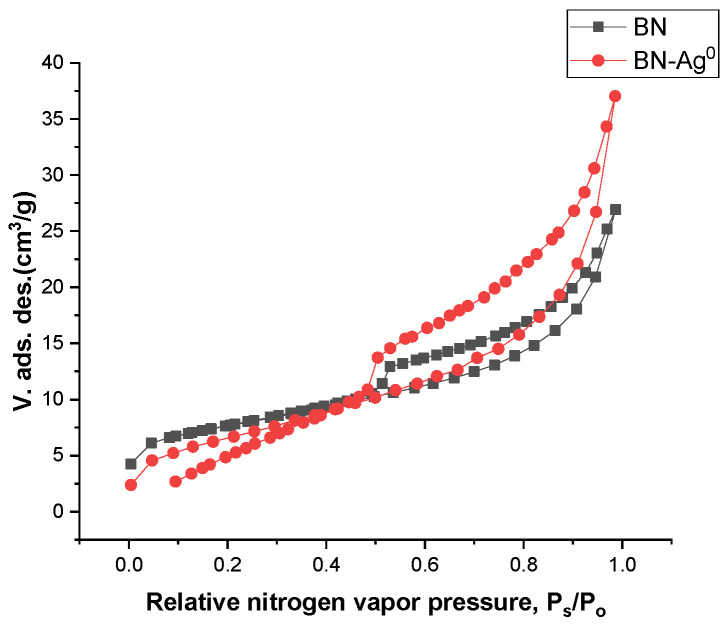
Adsorption–desorption isotherms of BN and BN-Ag^0^.

**Figure 5 molecules-27-05791-f005:**
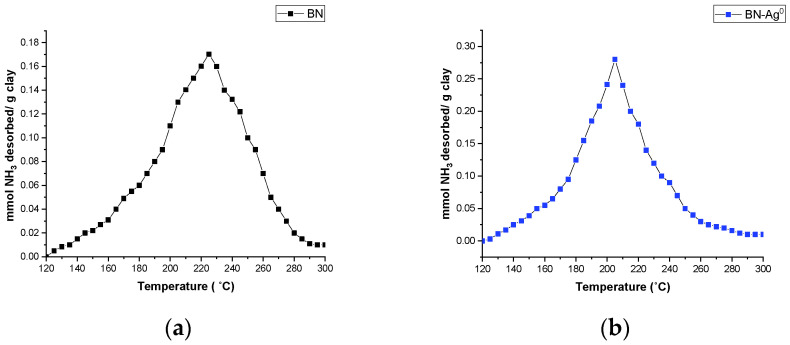
NH_3_ TPD diagram of temperature programmed desorption for BN (**a**) and BN-Ag^0^ (**b**).

**Figure 6 molecules-27-05791-f006:**
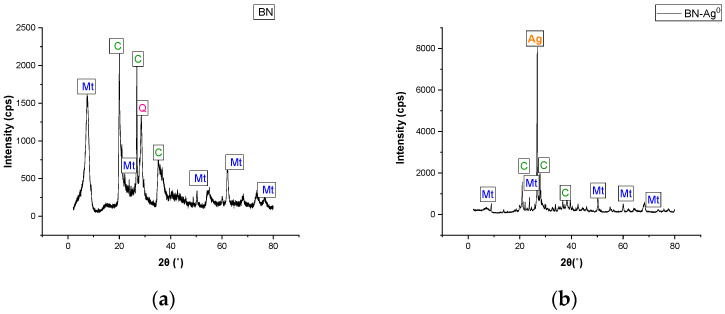
XRD diffractograms of BN (**a**) and BN-Ag^0^ (**b**).

**Figure 7 molecules-27-05791-f007:**
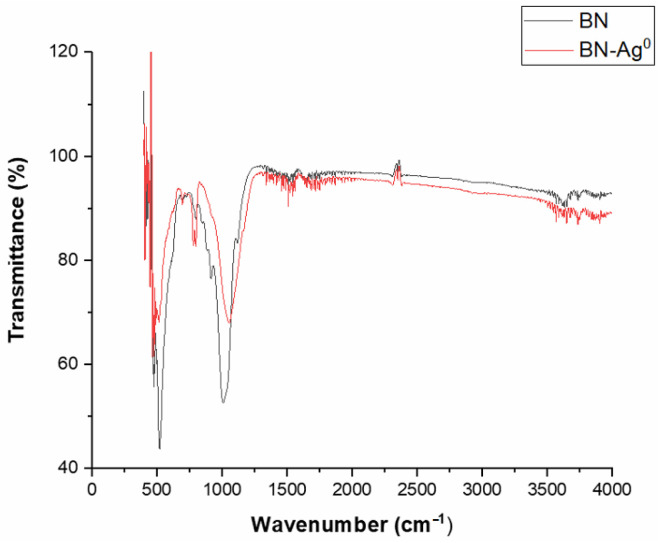
FTIR spectra of BN and of BN-Ag^0^.

**Figure 8 molecules-27-05791-f008:**
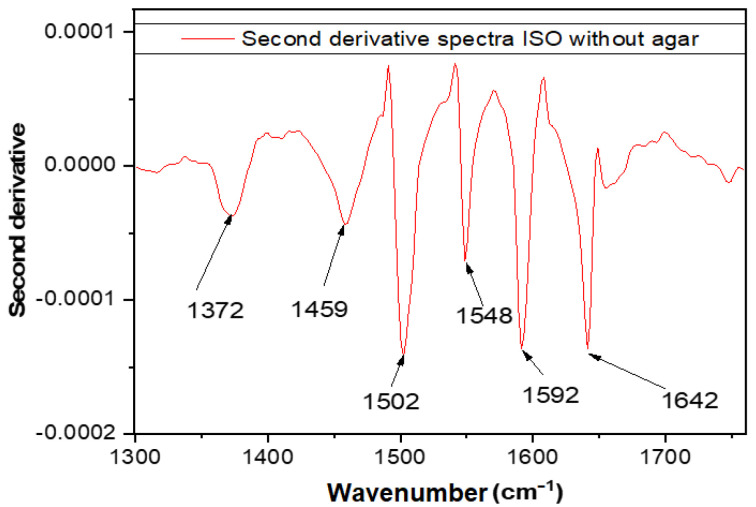
The ATR-IR absorption spectrum attributed to the 1300–1800 cm^−1^ region for the *ISO SS* strain from which the agar-agar spectrum was extracted [28].

**Figure 9 molecules-27-05791-f009:**
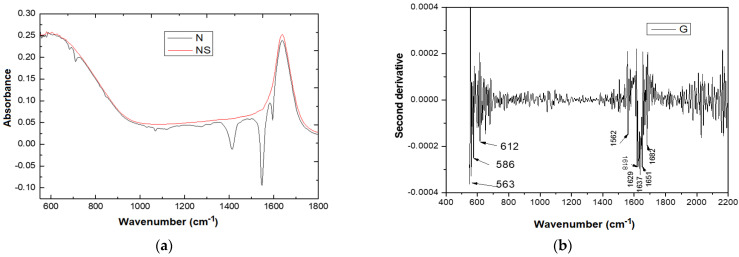
Spectra of absorption in ATR based on the wave number for anaerobically stabilized (N) sludge samples and anaerobic autoclaving (NS) treated anaerobic treatment sludge (NS) (**a**); A2-α derivative of the ATR spectrum according to the wave number for the Nutrient (**b**), N (**c**) and NS (**d**) samples [28].

**Figure 10 molecules-27-05791-f010:**
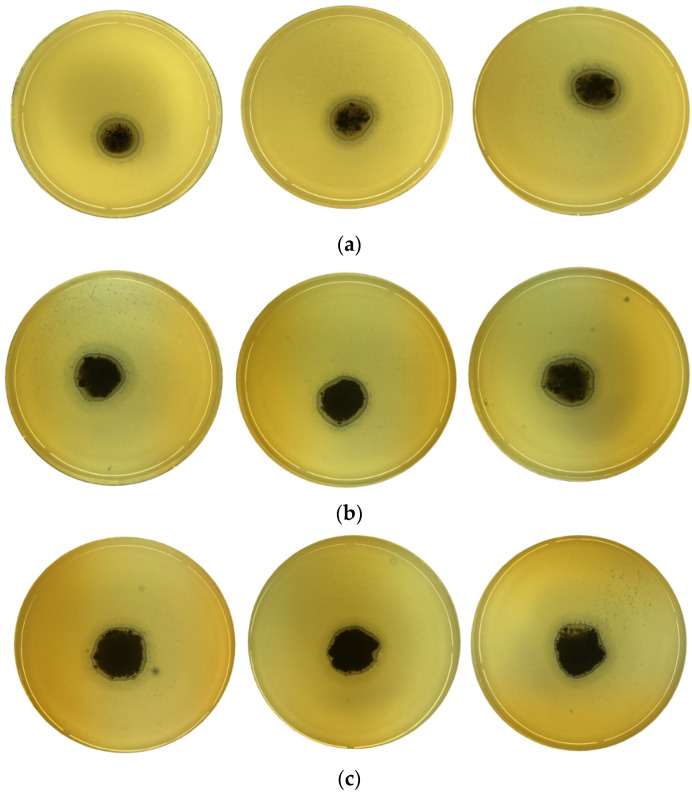
Inhibition zone revealing the antibacterial activity at 37 °C of different BN-Ag^0^ catalyst doses: 10 mg (**a**), 20 mg (**b**) and 30 mg (**c**), (in triplicate) for *ISO SS* strain.

**Figure 11 molecules-27-05791-f011:**
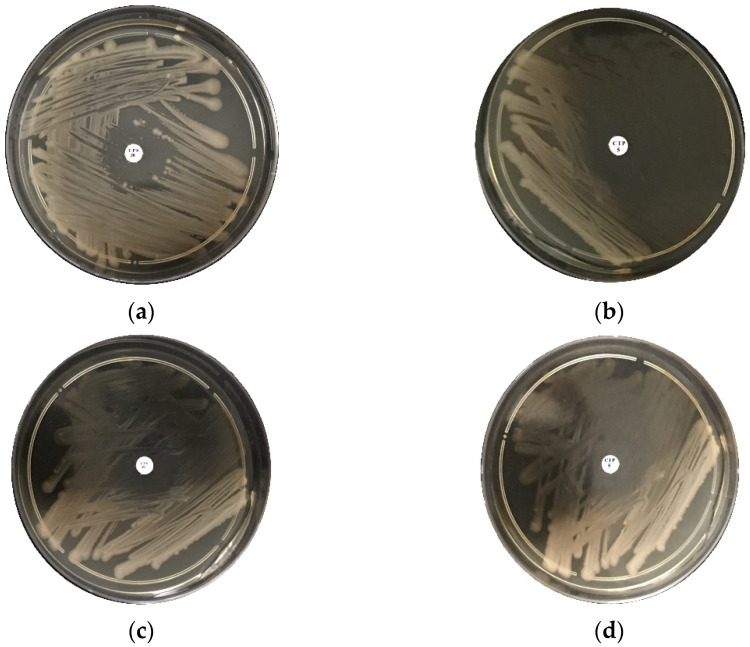
Antimicrobial behavior of ceftaroline fosamil CPS 30 (**a**), ciprofloxacin CIP 5 (**b**) against ISO SS and ceftaroline fosamil CPS 30 (**c**), ciprofloxacin CIP 5 (**d**) against *E. coli*.

**Figure 12 molecules-27-05791-f012:**
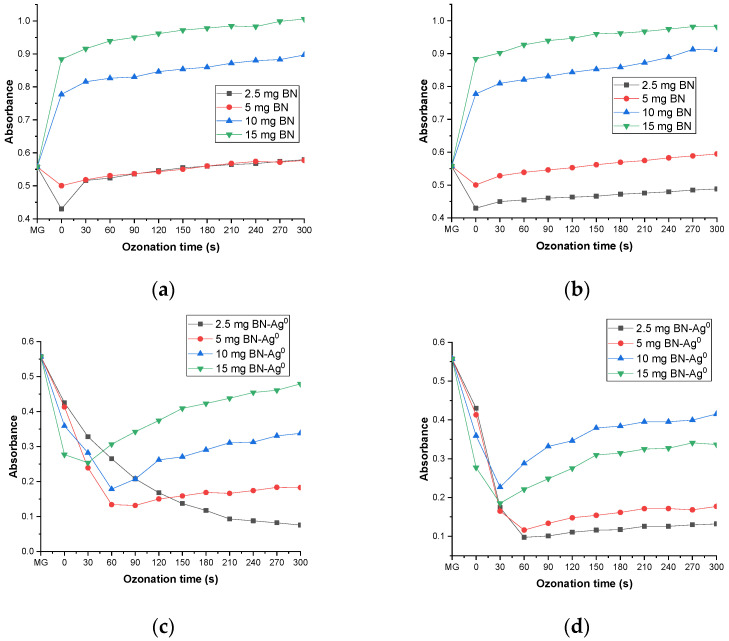
Catalytic ozonation of MG solution (5 × 10^−5^ M, initial absorbance 0.5572), using different amounts of BN (2.5, 5, 10 and 15 mg) and different doses of O_3_, 0.5 g∙h^−1^ (**a**) and 1 g∙h^−1^ (**b**); different amounts of BN-Ag^0^ (2.5, 5, 10 and 15 mg) and different doses of O_3_: 0.5 g∙h^−1^ (**c**) and 1 g∙h^−1^ (**d**), 20 mL of dye solution, ozonation time (0–300 s).

**Figure 13 molecules-27-05791-f013:**
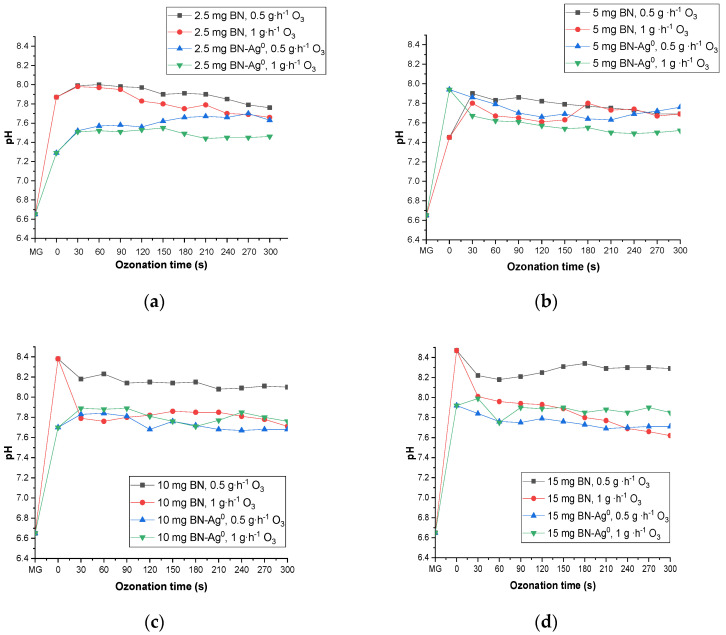
The influence of ozonation time on the pH of the MG solution (20 mL, initial pH = 6.65) ozonated with different amounts of BN and BN-Ag^0^: 2.5 mg (**a**), 5 mg (**b**), 10 mg (**c**), 15 mg (**d**); 0.5 and 1 g∙h^−1^ O_3_; ozonation time (0–300 s).

**Figure 14 molecules-27-05791-f014:**
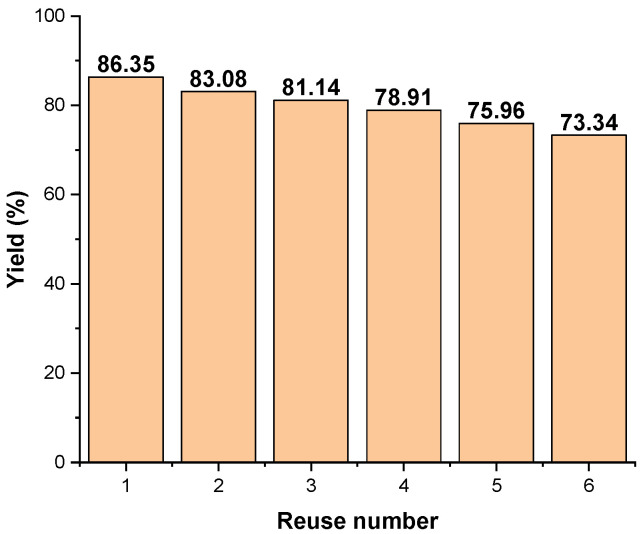
Reuse of BN-Ag^0^ catalyst in the catalytic destruction of industrial pollutant MG (20 mL). Destruction conditions: concentration MB = 5 × 10^−5^ M, mass of catalyst 2.5 mg, 0.5 g ∙ h^−1^ O_3_, ozonation time 300 s.

**Table 1 molecules-27-05791-t001:** Mineralogical composition of BN.

No.crt.	Component	Content (%)
1	Montmorillonite	62
2	Cristobalite	35
3	Quartz	3
4	Illite	-
5	Feldspars	-

**Table 2 molecules-27-05791-t002:** Textural results of BN and BN-Ag^0^.

Sample Name	Specific Surface Area (m^2^/g)	Total Pore Volume (cm^3^/g)	Average Pore Diameter (nm)	Total Surface Acidity × 10^3^ (Moles of Desorbed NH_3_/g Clay)
BN	25.70	0.0316	6.356	0.50
BN-Ag^0^	24.19	0.0473	9.332	0.65

**Table 3 molecules-27-05791-t003:** Quantitative analysis of BN-Ag^0^ obtained by XRD.

No.crt.	Phase Name	Formula	Content (%)
1	Montmorillonite	Ca_0.06_Na_0.21_K_0.27_Al_1.64_	61 (13)
2	Talc	Mg_3_(OH)_2_Si_4_O_10_	11 (3)
3	Silver	Ag	0.47 (19)
4	Quartz	SiO_2_	18 (4)
5	Muscovite	KAl_2_(AlSi_3_O_10_)(OH)_2_	9 (3)

**Table 4 molecules-27-05791-t004:** *ISO SS* strain characterization.

Experimental Sample	Sample Code Isolated	Gram Staining	Oxidase Test	Cultural Features on Nutrition Gel
Untreated sludge	*ISO SS*	G-	OXI-	coccobacillus, small colonies, round edges, glossy appearance, fecaloid smell

**Table 5 molecules-27-05791-t005:** Positioning and assignment of the bands presented in the ATR-IR spectra characteristic of the *ISO SS* strain from which the agar-agar spectrum was extracted [28].

Wavelength (cm^−1^).	Assignment	Reference
1372	Deformation of N-H, C-H bonds	[29]
1459	Asymmetric CH_3_ bending modes of the methyl groups of proteins	[30]
1502	Amide II (an N-H bending vibration coupled to C-N stretching)	[29]
1548	Amide II of proteins	[30]
1592	C=N, NH_2_ adenine	[29]
1642	Amide I band of protein	[29,31]

**Table 6 molecules-27-05791-t006:** Attribution of the main FTIR absorption bands for nutrient agar medium (G), anaerobic stabilized treatment sludge (N) and anaerobically stabilized treatment sludge treated by autoclaving (NS) [34].

The Main Absorption Bands FTIR (cm^−1^) for Different Samples			Assignment	Reference
G	G	NS		
563	563	563	CH deformation vibrations outside the plane	[35]
	572	572		
612				
		640		
		670		
	702			
	1538		Amide II and secondary amines	[29]
1562	1562	1562		
	1589		C=C aromatic skeleton	
	1599			
1681		1619	C-C stretching of the phenyl groups	
1629			Amide I, carboxylates, alkenes	[35]
1637		1635		
1661		1652		
1682		1686		

**Table 7 molecules-27-05791-t007:** Zone of inhibition (mm) of each sample (in triplicate) for *ISO SS* strain and *E. coli*.

Sample	BN-Ag^0^ Amount (mg)	Inhibition Zone (mm)
*ISO SS*	10	3
	20	3.5
	30	4
*E. coli*	10	2.3
	20	3.6
	30	4

## Data Availability

Not applicable.

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
