# Peer review of "Silver Nanoparticles Incorporated on Natural Clay as an Inhibitor against the New ISO SS Bacteria Isolated from Sewage Sludge, Involved in Malachite Green Dye Oxidation"

_molecules, 2022, doi:10.3390/molecules27185791_

Round 1
Reviewer 1 Report
The writing of this mauscript is very rough. Authors are suggested to revise it from the viewpoints below.
1. In the Introduction part, more discussion on relevant previsous works is needed to reveal the novelty and importance of this work.
2. Details on the method used to prepare the Ag-clay composite should be given at the beggining of Results and DIscussion part.
3. The title doesn't cover the content, as the catalytic property of the composite is also studied in details.
4. Micromophology of the samples needs to be studied. This is very important. Authors state that "...... core shell material.", but no evidence is provided.
5. The quality of figures needs to be improved.
6. Abbrevation should be explained correctly when it appears for the first time.
Author Response
The writing of this manuscript is very rough. Authors are suggested to revise it from the viewpoints below.
Q1: In the Introduction part, more discussion on relevant previsous works is needed to reveal the novelty and importance of this work.
A1. In the introduction, the following informations were entered:
The presence of dyes in the environment has a negative impact on biodiversity and especially on aquatic fauna. This should impose a stricter regulation on industrial producers which removes dye-rich wastewater. The special interest dedicated to organic dyes results from their much higher concentration compared to a wide variety of organic pollutants released in wastewater from different industries (food, pharmaceutical, textile etc.), hospital effluents and others.
MG (also known as Diamond Green B or Victoria Green B) is classified in the dye industry as a triarylmethane dye. It is usually used in agriculture and the fish industry as a powerful anti-bacterial, anti-fungal, anti-parasitic and dye agent, but it is also used for paper, leather, silk and other materials. The use of MG has been forbidden in several countries as it is not approved by the US Food and Drug Administration, but this dye is still widely used due to its low cost, easy availability and efficacy. MG possibly causes carcinogenic, mutagenic and teratogenic effects on humans, if it enters in the food chain. Also, it is highly cytotoxic to mammalian cells and acts as a tumor enhancing agent.
The reason for using natural bentonite BN from Valea Chioarului, MaramureÈ™ (Romania), as an active phase support and as an alternative to activated carbon, zeolites or other materials, was due to its natural wealth of the country, good adsorption properties and very low exploitation price. Mainly, we wanted to capitalize the indigene natural clay.
Q2: Details on the method used to prepare the Ag-clay composite should be given at the beggining of Results and Discussion part.
A2: Experimental protocol for the synthesis of BN-Ag0 are now given at the beginning of Results and Discussion part.
Q3: The title doesn't cover the content, as the catalytic property of the composite is also studied in details.
A3: The title of this manuscript has been changed from “Silver nanoparticles embedded cationic clay as an efficient inhibitor against the new ISO SS bacteria isolated from sewage sludge" with "Silver nanoparticles incorporated on natural clay as inhibitor against the new ISO SS bacteria isolated from sewage sludge, involved in Malachite Green dye oxidation".
Q4: Micromophology of the samples needs to be studied. This is very important. Authors state that "...... core shell material.", but no evidence is provided.
A4: Following the in-depth analysis of the specialized literature on core shell material, it was decided to abandon this term for the BN-Ag0 nanomaterial.
However, Table 3 was added to the manuscript, in which the mineralogical composition of bentonite modified with silver is presented at the line 149.
Q5: The quality of figures needs to be improved.
A5: The quality of the figures presented in the manuscript has been improved.
Q6: Abbrevation should be explained correctly when it appears for the first time.
A6: Abbreviation are now explained correctly when it appears for the first time.

Reviewer 2 Report
The submitted manuscript shows the antibacterial activity of bentonite modified with Ag particles against ISO SS, a Gram-negative bacterium. The same material was also used in the degradation of the dye - malachite green (MG) - in the ozonation process. The material is not very original both in terms of the type of active phase and support. Below I present various comments that may improve the quality of this manuscript, but at the beginning I would like to draw attention to two issues: (i) talking about the produced material as a core-shell structure is a serious abuse (I suggest referring to the literature describing the core-shell structure to see what the difference is) and (ii) it is necessary to explain why it was decided to use bentonite as the active phase support, in principle it could be replaced by any other material that would increase the dispersion of Ag particles.
1. The abbreviation BN used in the abstract should be explained there. Continuing with this, the abbreviation BN has been defined twice in the main text: (i) in line 75 in Introduction as "silica material" and (ii) ln line 250 in Materials and Methods as "bentonite". Assuming that BN is bentonite, the definition introduced in line 75 should be clarified. Bentonite is a layered aluminosilicate (usually called clay), and not, as imprecisely explained there, a silica material.
2. Impregnation (line 259) is also imprecise in relation to ion exchange to which the starting clay was subjected. It should be remembered that the impregnation can actually be carried out by dry or wet procedure, but an introduced modifier (i.e. impregnating solution) remains in the modified material and only a solvent is removed by evaporation.
3. Ag ions exist in AgNO3 on oxidation state +1, not +2 as erroneously indicated in line 266.
4. The sentences contained in lines 305-310 are completely unnecessary and do not add anything to the work. The study of porosity of the materials was carried out using low-temperature adsorption of nitrogen as an adsorbate. Therefore, the description of this method should be limited to N2.
5. However, I would like to learn more about the methodology of measuring temperature-programmed ammonia desorption. How were the samples prepared for measurement? At what temperature was the adsorption carried out? How were the physically adsorbed forms removed prior to desorption? What were the purging gas flows? What was the mass of the material tested? What was used as a signal detector?
6. The manuscript does not present the assumed or actual content of Ag nanoparticles in the modified bentonite, which is very important information from the point of view of the discussion of the efficiency of the tested material, as well as the possible reproducibility of the synthesis.
7. It should be remembered that experimental values should be presented with an accuracy that corresponds to potential errors of a method. And so, the accuracy of specific surface area should be reduced to 0.1 m2/g, total pore volume - 0.01 cm3/g, average pore diameter - 0.1 nm, and yield of MG degradation - 0.1%.
8. Having measured NH3-TPD profiles, they should be shown, as well as deconvolved, trying to discuss types of acid sites present on the surface. Taking into account the very short information on the obtained results given on page 4, I have the impression that the authors forget about possibility of binding NH3 molecules with acid Lewis sites.
9. The method of X-ray diffraction is usually denoted by the acronym XRD and for the sake of clarity I propose to use one in the manuscript.
10. Is it known what is the mass content of the montmorillonite phase in the studied bentonite from the Chioarului Valley, MaramureÈ™ County, Romania? The diffraction patterns shown in Fig. 3 suggest the presence of considerable amounts of impurities, especially of cristobalite.
11. It was found that during the ozonation of MG, pH changed that may affect the rate of the reaction. Have the authors checked the effect of pH on the MG conversion by perfoming the reaction at a controlled pH?
12. In repeated MG ozonation tests, a gradual decrease in the activity of the catalyst can be seen. Has this effect been explained, e.g. by examining the content and form of the active Ag phase after the reaction?
Author Response
Dear Reviewer,
In this document we have answered your questions and suggestions.
The submitted manuscript shows the antibacterial activity of bentonite modified with Ag particles against ISO SS, a Gram-negative bacterium. The same material was also used in the degradation of the dye - malachite green (MG) - in the ozonation process. The material is not very original both in terms of the type of active phase and support. Below I present various comments that may improve the quality of this manuscript, but at the beginning I would like to draw attention to two issues: (i) talking about the produced material as a core-shell structure is a serious abuse (I suggest referring to the literature describing the core-shell structure to see what the difference is) and (ii) it is necessary to explain why it was decided to use bentonite as the active phase support, in principle it could be replaced by any other material that would increase the dispersion of Ag particles.
- Following the in-depth analysis of the specialized literature on core shell material, it was decided to abandon this term for the BN-Ag0
- The reason for using natural bentonite BN, Valea Chioarului, Maramures as an active phase support and as an alternative to activated carbon, zeolites or other materials, was due to its availability in nature, good adsorption properties and very low exploitation price. And mainly it was wanted to capitalize the local natural clay.
Q1: The abbreviation BN used in the abstract should be explained there. Continuing with this, the abbreviation BN has been defined twice in the main text: (i) in line 75 in Introduction as "silica material" and (ii) ln line 250 in Materials and Methods as "bentonite". Assuming that BN is bentonite, the definition introduced in line 75 should be clarified. Bentonite is a layered aluminosilicate (usually called clay), and not, as imprecisely explained there, a silica material.
A1: BN is now explained in abstract. Because in the manuscript I intervened with explanations regarding the abbreviations used, explanations regarding the use of the original bentonite from Romania and additional informations, line 75 is now line 96 and line 250 became line 344.
The definitions have been clarified and the term bentonite is now used throughout the manuscript.
Q2: Impregnation (line 259) is also imprecise in relation to ion exchange to which the starting clay was subjected. It should be remembered that the impregnation can actually be carried out by dry or wet procedure, but an introduced modifier (i.e. impregnating solution) remains in the modified material and only a solvent is removed by evaporation.
A2: Line 259 is now line 105. We clarified the work protocol and we hope that now the working method used is more precise.
Q3: Ag ions exist in AgNO3 on oxidation state +1, not +2 as erroneously indicated in line 266.
A3: Line 266 is now line 117. The erroneous indication of the valence of silver has been modified.
Q4: The sentences contained in lines 305-310 are completely unnecessary and do not add anything to the work. The study of porosity of the materials was carried out using low-temperature adsorption of nitrogen as an adsorbate. Therefore, the description of this method should be limited to N2.
A4: The informations presented in lines 305-310 were removed from the manuscript.
Q5: However, I would like to learn more about the methodology of measuring temperature-programmed ammonia desorption. How were the samples prepared for measurement? At what temperature was the adsorption carried out? How were the physically adsorbed forms removed prior to desorption? What were the purging gas flows? What was the mass of the material tested? What was used as a signal detector?
A5: Surface acidity and basicity was determined using temperature programmed desorption (TPD).
The diagram of the used plant is shown in Figure 1.
|
|
Figure 1. Scheme of the calcination plant and "in situ" determination of acid-base properties [1].
1-gas cylinder, 2-NaOH drying column, 3-reactor for TPD, 4-buffer container, 5-vapor driving container for NH3, 6-Hg manometer, 7-hermetic test tubes for gas absorption, 8-fluometer.
The speed of achieving the thermal regime was programmed to 2ËšC·min-1. Approximately 0.2 g of sample, was introduced into the reactor (3). A granulometric fraction of 0.02-0.2 mm particle diameter was selected from the solid sample.
A thermal regime previously programmed was followed. A scheme of the thermal regime is presented in Figs. 2 a,b [1-3].
|
(a) |
(b) |
|
Figure 2. The thermal profile used in the programmed thermo-desorption to determine the acid properties (a) and the basic properties (b) of the clay. |
|
|
- T1: ambient temperature, representing for the actual process, the starting temperature (20 ËšC); - T2: temperature of the bearing (400 ËšC); - T3: temperature at which ammonia is "injected", after which this temperature is maintained and washed with nitrogen in a 2 h purge; ammonia is introduced at 120 ËšC; - T4: temperature up to which thermal desorption is achieved (550 ËšC). |
- T1: ambient temperature, representing for the actual process, the starting temperature (20 ËšC); - T2: temperature of the bearing (400 ËšC); - T3: temperature at which carbon dioxide is "injected" and has a value of 80 ËšC, after which this temperature is maintained and washed with nitrogen in a 2 h purge; - T4: temperature up to which thermal desorption is achieved, i.e. 500-550 ËšC. |
The plant works under a nitrogen flow, the basic or acidic gas being introduced only during experiments to study the acidic or basic properties of the material. The plant is managed with the "software" that can program the temperature, the duration of maintaining the temperature, the increase or decrease of the temperature according to an increase slope, expressed in ËšC/min.
In our protocol, we performed a heat treatment at high temperature, and then we lowered the temperature to 120 ËšC, when we performed an injection for 5 min, with NH3, in the N2 carrier gas. The installation was continuously under nitrogen flow.
In order to remove excess ammonia, a nitrogen wash is carried out for about 2 h. In this way, we make sure that only physico-chemically bound ammonia remained in the pores of the studied material.
After purging, temperature programmed desorption (TPD) stage is occurring. Desorbed ammonia depends on the working temperature (when studying the acidity of clays). Desorbed ammonia is highlighted by indirect titration.
The gas desorbed during the controlled thermal desorption was bubbled into test tubes, containing the acid component in appropriate solutions. The determination of the acidic character of the clay was carried out by absorbing the gases resulting from the temperature programmed desorption, in test tubes containing 0.02 N H2SO4 solution (turn from violet to gray), followed by titration of the excess with a 0.01 N NaOH solution, in the presence the Tashiro’s indicator solution.
For the determination of basicity, the same protocol was followed, with the difference that CO2 was injected for 5 min. The washing with nitrogen flow took place for 2 h and it was passed to temperature programmed desorption (TPD). The basic character of the clay was determined by absorbing the gases resulting from the temperature programmed desorption, in test tubes containing 0.01 N NaOH solution, followed by the titration of the excess with 0.005 N H2SO4 solution, in the presence of the Tashiro’s indicator solution (turn from green to gray).
Q6: The manuscript does not present the assumed or actual content of Ag nanoparticles in the modified bentonite, which is very important information from the point of view of the discussion of the efficiency of the tested material, as well as the possible reproducibility of the synthesis.
A6: According to the XRD analysis, the actual content of Ag nanoparticles in the modified bentonite is 0.47%. I added the mineralogical composition of BN-Ag0 in the manuscript in Table 3, line 176.
Q7: It should be remembered that experimental values should be presented with an accuracy that corresponds to potential errors of a method. And so, the accuracy of specific surface area should be reduced to 0.1 m2/g, total pore volume - 0.01 cm3/g, average pore diameter - 0.1 nm, and yield of MG degradation - 0.1%.
A7: The new values from the manuscript are now presented with an accuracy that takes into account the potential errors of the methods used, as you indicated in your comment.
Q8: Having measured NH3-TPD profiles, they should be shown, as well as deconvolved, trying to discuss types of acid sites present on the surface. Taking into account the very short information on the obtained results given on page 4, I have the impression that the authors forget about possibility of binding NH3 molecules with acid Lewis sites.
A8: The NH3-TPD graphs are now shown in manuscript in Figure 3. The structural changes during preparation lead to the partial or total destruction of the octahedral layers in the crystalline structure, which immediately results in an increase in Lewis acidity - attributed to surface cations. During the preparation, in order to establish the balance at the charge level, the protons in the ion exchange solution replace the exchange interlamellar cations (especially the Na+ ions). These protons contribute to increasing the surface acidity. It is also possible that the protons in the marginal -OH groups of the octahedra become more labile due to structural deformations due to ion exchange, which also leads to an increase in Brønsted acidity. In the case of the adsorption of basic gases (ammonia), it is important that the adsorbent has a surface acidity as high as possible for the best efficiency of the adsorption process.
Q9: The method of X-ray diffraction is usually denoted by the acronym XRD and for the sake of clarity I propose to use one in the manuscript.
A9: The method of X-ray diffraction is now denoted in manuscript by the acronym XRD.
Q10: Is it known what is the mass content of the montmorillonite phase in the studied bentonite from the Chioarului Valley, Maramures County, Romania? The diffraction patterns shown in Fig. 3 suggest the presence of considerable amounts of impurities, especially of cristobalite.
A10: The mass content of montmorillonite phase of the studied bentonite is 62%. We have added the mineralogical composition of natural bentonite in the manuscript, in Table 1 on line 127. The diffraction patterns shown in Fig. 3 became Fig. 4, line 168.
Q11: It was found that during the ozonation of MG, pH changed that may affect the rate of the reaction. Have the authors checked the effect of pH on the MG conversion by performing the reaction at a controlled pH?
A11: We can talk about the catalytic ozonation when the efficiency of ozonation in the presence of a catalyst is higher than without it at the same pH values. However, when investigating catalytic activity of a solid catalyst, also another condition must be fulfilled, the total effect of catalytic ozonation must be higher than a combined effect of adsorption on the catalyst surface and ozonation alone at the same pH values. As we can see the measurements of pH in both ozonation alone and catalytic ozonation of MG are basic requirements which allow a separation of two effects: decomposition of ozone (and perhaps formation of OH• radicals which is highly pH dependent) and catalytic activity. During the ozonation of MG, pH changes may affect the rate of the reaction. Also, it seems that some of the controversies in catalytic ozonation presented in the specialized literature can have been caused by the lack of proper pH control.
When the experiment concerns the catalytic decomposition of ozone, the results must be compared with the decay of ozone in water at the same pH. Ozone decomposition strongly depends on pH. It seems to be easy to assure the same pH in both experiments but usually is quite hard to control pH after introducing the catalyst into dye solution (or dye solution with corrected pH e.g. with acid or pH stabilized with buffer). Introducing the catalyst (BN or BN-Ag0) into dye solution dramatically change the pH value (from 6.5 to 8.5 and 6.5 to 8, respectively). There are several reasons for that, of which the most important is the presence of mineral impurities in the catalyst (cristobalite and quartz). Before starting the catalytic ozonation process, the pH of each MG solution was adjusted to 6.65 using a 0.2 M HCl solution.
The ozonation process of MG (5∙10-5 M) was already published in a previous work [4].
Q12: In repeated MG ozonation tests, a gradual decrease in the activity of the catalyst can be seen. Has this effect been explained, e.g. by examining the content and form of the active Ag phase after the
A12: In the repeated ozonation tests of MG, it was found that the catalytic activity decreases slightly. This fact is possibly due to the distortion of the active sites of the material. Also, another possible factor that contributes to the decrease in catalytic activity is given by the transformations undergone by Ag+ ions during ozonation.
- Nistor, D., et al., Etude par desorption thermique programmee des proprietes des argiles modifiees. Journal of thermal analysis and calorimetry, 2004. 76(3): p. 913-920.
- Azzouz, A., et al., Assessment of acid–base strength distribution of ion-exchanged montmorillonites through NH3 and CO2-TPD measurements. Thermochimica Acta, 2006. 449(1-2): p. 27-34.
- Nistor, D., et al., Optimized procedure for clay pillaring with aluminum species used in depollution. Journal of thermal analysis and calorimetry, 2006. 84(2): p. 527-530.
- Mirilă, D.-C., et al., Organic Dye Ozonation Catalyzed by Chemically Modified Montmorillonite K10– Role of Surface Basicity and Hydrophilic Character. Ozone: Science & Engineering, 2020: p. 1-14.

Round 2
Reviewer 1 Report
Authors have addressed my concerns. The manuscript can now be accepted for publication.
Author Response
Thank you for your kind evaluation.
Reviewer 2 Report
Most of my comments have been taken into account, but there are still two issues:
1. Information on the content of Ag still not appeared in the revised manuscript. Moreover, the XRD technique is not a suitable tool for determining the Ag content [wt.%].
2. The values are still given with too much accuracy. As I suggested previously, the accuracy of specific surface area should be reduced to 0.1 m2/g, total pore volume - 0.01 cm3/g, average pore diameter - 0.1 nm, and yield of MG degradation - 0.1%.
Author Response
Q1. Information on the content of Ag still not appeared in the revised manuscript. Moreover, the XRD technique is not a suitable tool for determining the Ag content [wt.%].
A1. We performed SEM-EDX analysis to determine the morphology and incorporation of silver ions, shown in Figs 1 and 2.
Q2. The values are still given with too much accuracy. As I suggested previously, the accuracy of specific surface area should be reduced to 0.1 m2/g, total pore volume - 0.01 cm3/g, average pore diameter - 0.1 nm, and yield of MG degradation - 0.1%.
A2. The values were modified as you suggested at the first revision (highlighted in pages 6, 12, 13, 14, 15 and 17).
Thank you for your kind evaluation, precious comments and suggestions.